# Explaining Latent Representations of Neural Networks with Archetypal Analysis

Anna Emilie J. Wedenborg* †1, Teresa Dorszewski*1, Lars Kai Hansen1, Kristoffer K. Wickstrøm2, and Morten Mørup1

1Technical University of Denmark
2Arctic University of Norway
{aejew, tksc, lkai, mmor}@dtu.dk
{kristoffer.k.wickstrom}@uit.no

## Abstract

We apply Archetypal Analysis to the latent spaces of trained neural networks, offering interpretable explanations of feature representations of neural networks without relying on user-defined corpora. Through layer-wise analyses of convolutional networks and vision transformers across multiple classification tasks, we demonstrate that archetypes are robust, dataset-independent, and provide intuitive insights into how models encode and transform information from layer to layer. Our approach enables global insights by characterizing the unique structure of the latent representation space of each layer, while also offering localized explanations of individual decisions as convex combinations of extreme points (i.e., archetypes).

## 1 Introduction

With the growing adoption of deep learning in computer vision, the explainability of these models has become increasingly critical. The lack of transparency in their decision-making processes raises concerns about trust, accountability, and fairness-concerns that can potentially be mitigated through explainability methods that offer insights into how models process information and arrive at their predictions [1]. Although much of the existing work on explainability focuses on input-output relationships, such as saliency maps or class activation mappings [2, 3], there is a growing recognition that a deeper understanding of latent representations is essential [1, 4]. These internal feature spaces, formed across layers of a model, encode abstract and hierarchical information that underpins the model's reasoning process. By probing and interpreting these latent spaces, we can improve our understanding of information processing, uncover biases, and guide model debugging and improvements [5].

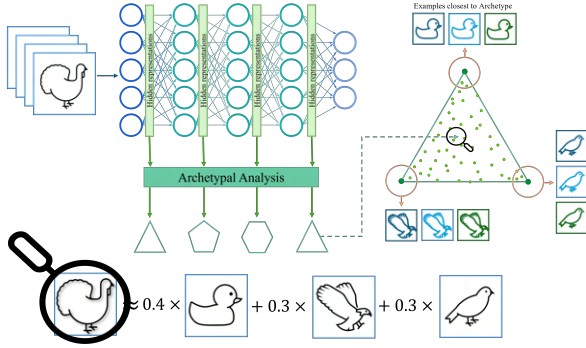

**Figure 1.** Our main analysis pipeline consists of 3 main steps: 1. Extraction of latent representations of the data after each layer, 2. Archetypal analysis of latent representations, where unique archetypes are learned for each layer independently, and 3. Analysis of the resulting simplex by projection of images/data onto the learned simplex and visualization of the closest data points.

Previous works have explored the latent space globally by investigating how concepts evolve across layers in convolutional neural networks (CNNs) [6] and vision transformers [7, 8], highlighting how vision models learn simple concepts (e.g. colors) in early layers and build up an understanding of more complex concepts (e.g. objects) in later layers. Crabbe et al. [9] introduced the SimplEX method to explain the latent space using a user-defined corpus, where each point is represented as a convex combination of corpus examples. This approach enables local explanations through feature attributions from individual data points.

Several recent works have explored interpretable representation learning via prototype or clustering mechanisms (e.g., [10–13]) These methods typically learn centroids or prototypes jointly with the network, thereby shaping the representation space through supervised or self-supervised optimization. In contrast, our framework applies Archetypal Analysis (AA) post-hoc to pre-trained latent spaces. While centroid-based methods represent data in terms of mean prototypes, AA decomposes the feature space into extreme points (archetypes) and expresses each

---

*Equal contribution.

†This work was supported by the Danish Data Science Academy, which is funded by the Novo Nordisk Foundation (NNF21SA0069429) and VILLUM FONDEN (40516), and the Novo Nordisk Foundation grant NNF22OC0076907 "Cognitive spaces-Next generation explainability".

Proceedings of the 7th Northern Lights Deep Learning Conference (NLDL), PMLR 307, 2026.

data point as a convex combination of these. As discussed by [14] and [15], this distinction reflects two fundamentally different philosophies: clustering seeks prototypical averages, whereas AA seeks distinct extremes, leading to interpretable geometries that capture the diversity of the learned features rather than their central tendency.

Sparse Autoencoders (SAEs [16]) and their variants (e.g., A-SAE [17]) have recently emerged as powerful tools for interpretable representation learning. These models achieve interpretability by introducing sparsity and modularity constraints directly during training, thereby shaping the latent space to yield disentangled components. Our approach, in contrast, provides a post-hoc geometric perspective: Archetypal Analysis (AA) decomposes pre-existing latent representations into convex combinations of extreme points (archetypes), without modifying or retraining the underlying model. This makes AA complementary to SAE-based approaches; while SAEs learn interpretable features by construction, AA reveals interpretable structure inherent in already trained models.

In this paper, we present a novel take on explainability of latent representations by examining what insights can be gained from analyzing latent representations of neural networks with Archetypal Analysis (AA) [18]. With this approach, we can provide an easy interpretable characterization of the latent space within each of the network layers in terms of distinct global characteristics. The proposed pipeline is illustrated in Figure 1.

Notably, AA is especially known for its easily interpretable results in which each data point is described as a convex combination of archetypes. The archetypes are in turn defined as convex combinations of the entire data (i.e., reside on the convex hull) and form distinct characteristics. The polytope formed by AA is also denoted the principal convex hull of the data [15]. Consequently, AA seeks to explain the data in terms of extreme representations as opposed to clustering that represents the data in terms of centroids. This corresponds to projecting the data onto a lower-dimensional simplex, which is spanned by extremes in the data. This framework has previously been extended to deep learning, but this has been based on regularization towards a polytope within the specific bottleneck layer of a (variational) auto-encoder representation and not as a means to characterize the structure of each layer of the network [19–22].

We show that AA can be applied to the latent space of deep learning models after they have been trained, thereby providing a post-hoc analysis approach for explaining how the model processes information at each step. We show that the derived archetypes are stable and independent of the used probing dataset. Whereas Crabbe et al. [9] explain

latent representation by convex combinations of a corpus of examples, we can explain latent representations by convex combinations of a small number of archetypes, which are data corpus independent and account optimally for the entire latent space. Therefore, our corpus (archetypes) is not selected by the user, but optimally found by the model during optimization and through analysis shown to be both data independent and robust. These explanations through extreme points also align more closely with how people reason from data, where more extreme category representatives produce stronger inductive inference [23, 24].

In summary, we introduce a post-hoc analysis framework for explainable AI that enables systematic, layer-wise analysis of neural representation spaces based on AA. The main contributions of this work are:

- A corpus-independent methodology applicable to any neural network architecture.

- Dataset-agnostic global explanations defined by the distinct characteristics of each layer's latent representation forming the archetypes.

- Easy interpretable observation specific local explanations expressed in terms of convex combinations of the extracted archetypes.

In addition, we empirically demonstrate the robustness and versatility of the proposed framework across diverse datasets and model architectures, highlighting both its reliability and interpretability. Through an in-depth layer-wise analysis of two vision transformers and a convolutional neural network fine-tuned on distinct classification tasks, we show how our method reveals broad structural patterns in the evolution of latent representations across layers, while also providing local, example-based explanations within an interpretable archetypal space.

## 2 Methodology

Here we present the fundamental theory of AA, followed by our proposed analysis pipeline for latent spaces using AA.

### 2.1 Archetypal Analysis

Archetypal Analysis aims to find the optimal reconstruction of the data, $\mathbf{X} \in \mathbb{R}^{M \times N}$ [18], by solving the following optimization objective, with $\mathbf{C} \in \mathbb{R}^{N \times K}$ and $\mathbf{S} \in \mathbb{R}^{K \times N}$,

$$\begin{aligned} \underset{\mathbf{C},\mathbf{S}}{\text{minimize}} \quad & \|\mathbf{X} - \mathbf{R}\|_F^2 \\ \text{subject to} \quad & \mathbf{R} = \mathbf{X}\mathbf{C}\mathbf{S} \\ & c_{n,k} \geq 0, \quad \sum_n c_{n,k} = 1, \\ & s_{k,n} \geq 0, \quad \sum_k s_{k,n} = 1, \end{aligned} \quad (1)$$

where $\mathbf{XC}$ forms the archetypes and $\mathbf{S}$ is the data projected onto the archetypal simplex. To solve this problem, we use the efficient least-squares framework presented in [25].

## 2.2 Latent Space Archetypal Analysis

Figure 1 provides a visual overview of our proposed framework, which consists of 3 main steps:

1. Extraction of latent representations.

2. Archetypal analysis of latent representations.

3. Analysis of the resulting simplex.

First, for the transformer models (ViT, DINOv2), latent representations were extracted from a selected layer. Patch embeddings were aggregated by mean pooling to obtain a single feature vector per image. For the convolutional model (ResNet50), the mean across each channel was used to obtain a vector per image. All latent features were standardized (zero mean, unit variance) across images to enable stable and unbiased AA.

Second, the feature vectors are processed using an AA module, where we fit $K$ archetypes. The number of archetypes is a hyperparameter in the module that depends on the complexity of the model and the user's wishes for a lower-dimensional representation of the model. This process is repeated ten times. In our study, the optimal number of archetypes is determined as the smallest number that simultaneously achieves high explained variance and high stability.

Finally, we visualize the latent space in terms of the extreme representations, and extract the images that are closest to the archetypes by calculating the distance between $\mathbf{X}$ and the archetypes $\mathbf{Z}$.

# 3 Experimental Setup and Evaluation

This section details the datasets and models used in our latent-space AA experiments and explains how performance is evaluated.

## 3.1 Models and Data

We run all experiments on three different vision models (two transformer-based and one convolutional network), namely ViT [26], DINOv2 [27] and ResNet50 [28]. We test two different datasets, Caltech-UCSD Birds-200-2011 Dataset [29] (CUB, 200 classes) and a dataset of the MedMNIST collection [30], (organCMNIST, 11 classes) [31]. All models are fine-tuned to perform classification on the datasets and reach competitive performance (model, dataset, and training details are in Appendix A). For CUB we extract 2-100 archetypes due to the high number of classes, while we extract 2-20 archetypes for organCMNIST. For ViT and DINOv2 we extract the latent representation after each of the twelve transformer blocks (all with $M = 768$), for ResNet50 we extract them after each of the five convolutional blocks ($M = 64, 128, 256, 512$).

## 3.2 Model evaluation

We extracted the representations of the full training set, as well as five non-overlapping splits of the training set to test for dataset independence (all with ten random runs for each number of archetypes). We test the robustness across the runs using Normalized Mutual Information (NMI) [32]. We determine how much of the information in the latent space is kept by the Archetypal Analysis by calculating the variance explained (VE),

$$\text{VarianceExplained}(\mathbf{X}, \mathbf{R}) = 1 - \frac{\|\mathbf{X} - \mathbf{R}\|_F^2}{\|\mathbf{X}\|_F^2}. \quad (2)$$

As AA is a non-convex problem, we also examined the stability of the solution, repeating the experiment ten times and using the NMI between each unique pair of solutions, such that ten experiments result in 45 different NMI values.

$$\text{NMI}(\mathbf{S}, \mathbf{S}') = \frac{2I(\mathbf{S}, \mathbf{S}')}{I(\mathbf{S}, \mathbf{S}) + I(\mathbf{S}', \mathbf{S}')}$$

$$\text{where } I(\mathbf{S}, \mathbf{S}') = \sum_{kk'} p(k, k') \log \frac{p(k, k')}{p(k)p(k')}, \quad (3)$$

$$p(k, k') = \frac{1}{N} \sum_{n=1}^{N} s_{k,n} s'_{k',n}$$

This can measure the stability of the result by measuring the shared information between $\mathbf{S}$ and $\mathbf{S}'$, if these are identical, the NMI will be one.

We also evaluate the results based on how well the archetypal simplex projection corresponds to the data classes by evaluating NMI($\mathbf{S}, \mathbf{Y}_{class}$), where $\mathbf{Y}_{class}$) is the one-out-of-K encoded class labels of the images. To contextualize the class experiment we compared it against a randomly permuted class label, NMI($\mathbf{S}, \mathcal{P}(\mathbf{Y}_{class})$). We test the robustness of the archetypes by calculating the consistency of the archetypes between splits (with ten runs each), as described in [33]. The consistency is defined as a similarity score between the archetypes $\mathbf{Z} = \mathbf{XC}$ of two separate runs $\mathbf{Z}$ and $\mathbf{Z}'$:

$$sim(\mathbf{Z}, \mathbf{Z}') = 1 - \overline{d}^2 / \overline{\sigma}^2, \quad (4)$$

where $\overline{d}^2$ is the average squared distance between $\mathbf{Z}$ and $\mathbf{Z}'$ (after matching the archetypes by proximity) and $\sigma^2$ is the average variance across the $M$ features of $\mathbf{X}$. We test the robustness using five archetypes. While this number is not necessarily optimal in terms

of maximizing similarity scores, it reflects a deliberate trade-off between quantitative performance and qualitative interpretability. Fewer archetypes tend to produce cleaner, more distinct representations, which enhances the explainability of the analysis, particularly in visualizations and conceptual clarity. Consequently, our similarity scores should be interpreted as a lower bound, acknowledging that a higher number of archetypes might yield better scores but at the cost of reduced interpretability.

We also compare the archetypes across the layers (only for ViT and DINOv2, which have the same number of features in each layer), to investigate how much the archetypes change between layers. For these experiments, we only compare across all runs of the full set for five archetypes.

## 3.3 Accuracy on the Archetypal Simplex

To evaluate how well the model objective, in this case classification, aligns with the AA we apply a K-Nearest-Neighbor classifier (KNN) with five neighbors on the archetypal simplex, where we classify unseen projected test samples by their nearest neighbors on the simplex. This is then compared to a KNN directly on the latent representation. This will first show us how much of the models' predictive abilities are lost when compressed to a lower-dimensional space, and second provide insights into the models predictive abilities in earlier layers.

## 3.4 Test image projection

To interpret the resulting simplex, we projected five images onto the simplex and visualized each archetype using the five closest images, providing intuitive insight into the learned representations. The latent features ($\mathbf{F} \in \mathbb{R}^{M \times N}$) of $N$ data points are collected into $\mathbf{X} \in \mathbb{R}^{M \times N}$ and projected onto the simplex $\mathbf{S} \in \mathbb{R}^{K \times N}$ (eq. 1). The test samples are projected onto the archetypal simplex by solving the quadratic program described in [25] between the original archetypes, $\mathbf{Z}$, and the test samples, $\mathbf{X}^{test} \in \mathbb{R}^{M \times N_{test}}$.

## 3.5 Scalability

The AA procedure as implemented in [25] requires for the $\mathbf{S}$ update the computation of $\mathbf{C}^\top \mathbf{X}^\top \mathbf{X}$ which is $\mathcal{O}(NMK)$ and Hessian $(\mathbf{XC})^\top (\mathbf{XC})$ which is $\mathcal{O}(MK^2)$ whereas the sequential minimal optimization updates requires in the order of $K^2$ iterations for each of the $N$ columns of $\mathbf{S}$, i.e. $\mathcal{O}(NK^2)$ resulting in an overall complexity of $\mathcal{O}(NMK + K^2(M+N))$. The update of $\mathbf{C}$ is based on an active set procedure scaling in the size of the active set $|A| \ll N$ as $\mathcal{O}(K|A|^3)$. Typically, the size of the active set $|A|$

and the number of archetypes $K$ remain small, however, we note that in the case where either become large gradient based efficient optimization based on the PCHA algorithm [15] can be invoked with pr. iteration cost of $\mathcal{O}(NMK)$ whereas trivial parallellization of the associated matrix products can be implemented. Consequently, by use of suitable implementations of the AA procedure (see also [33] for an overview of optimization procedures) the method scales well and can be used for the post-hoc analysis of large datasets.

## 4 Results and Discussion

### 4.1 Latent Space Archetypal Analysis

We analyze the latent spaces of large vision models using Archetypal Analysis. Figure 2(a) illustrates the VE and NMI for organCMNIST based on layer one. Here, it can be observed that both the NMI and VE exhibit smooth trends as the number of archetypes increases, with stable solutions already emerging at low dimensions. Detailed results, including the optimal number of archetypes per model and dataset along with NMI and VE scores, are reported in Appendix B. Surprisingly, these high-dimensional embeddings (e.g., $M = 768$ for ViT and DINOv2) collapse onto a low-dimensional convex polytope that can be described with as few as three archetypes. Even with only a handful of archetypes, AA consistently explains a large fraction of the latent variance (80–99%) in both some of the early and later layers, the main exception arises in the final layers of CUB, where additional archetypes are needed to disentangle 200 fine-grained bird categories.

Determining the "right" number of archetypes remains an open question. Nevertheless, our empirical findings point to a surprisingly low intrinsic dimensionality of the latent spaces, particularly in early and intermediate layers. For visualization purposes we therefore standardize on five archetypes, which strike a balance between interpretability and expressiveness. Importantly, AA seeks to provide the best-fit simplex for the specified number of archetypes; the resulting explanations adapt to the chosen dimensionality, revealing the most extreme latent factors compatible with that constraint.

To examine the learned archetypes, we extract the five closest data points (images) to the archetype in archetypical space. For example, the first and last layer for ViT trained on CUB is shown in Figure 3. The archetypes for all layers, models and datasets can be found in Appendix C (all for five archetypes). A qualitative analysis of these archetypes reveals that in early layers the representations are spanned by archetypes relating to mainly background and color information and in later layers the network becomes more adapted to the task and the repre-

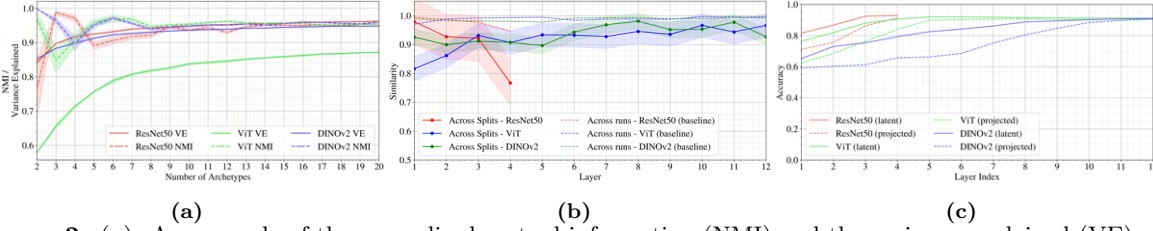

**Figure 2.** (**a**): An example of the normalised mutual information (NMI) and the variance explained (VE) as a function of the number of archetypes for layer one of the organCMNIST, the rest can be found in Appendix C. (**b**): The consistency of archetypes across different dataset splits for the CUB dataset. The consistency for the organCMNIST dataset can be found in Figure D.1. (**c**): A 5-nearest neighbours matching on the image label across different layers with twenty archetypes for the OrganCMNIST dataset.

sentations are more and more driven by the actual classes, in this case different species of birds. This also aligns with previous findings from studies that explored how concepts evolve in vision transformers and CNNs [6, 8].

We also examine archetypes with the CUB data set in the pretrained networks (Figure D.4 and D.5), to see if meaningful structures already emerge during pretrained. We find no clear archetypes, especially no class separation in late layers, which is due to the network not being trained for such a specialized task. We leave a more thorough investigation on general and bigger datasets for future work. Our method holds the potential to uncover emerging structures during pretraining and can lead to a deeper understanding of what these models learn and understand.

A key property of AA is that each data point can be expressed as a convex combination of the learned archetypes. This characteristic enables intuitive interpretation of the latent space. As illustrated in Figure 4, interpolations between pairs of archetypes yield images that are visually and semantically meaningful blends of the corresponding archetypal features. For instance, the image generated between the "light background" and "dark background" archetypes exhibits a clear mixture of a light background combined with a black frame. Similarly, the interpolation between "sky" and "grass" archetypes results in an image of leaves with visible sky shining through. These examples demonstrate how AA organizes data in the latent space in a way that is both interpretable and grounded in human-understandable visual semantics.

## 4.2 Data Independent and Robust Archetypes

While the archetypes are extracted from latent representations of a fixed dataset, our experiments show that the archetypes themselves are largely data-independent and reflect the intrinsic structure of the model's representational space. We rerun the archetypal analysis with varying datasets and find that the archetypes are stable across the different dataset splits. The example images closest to the

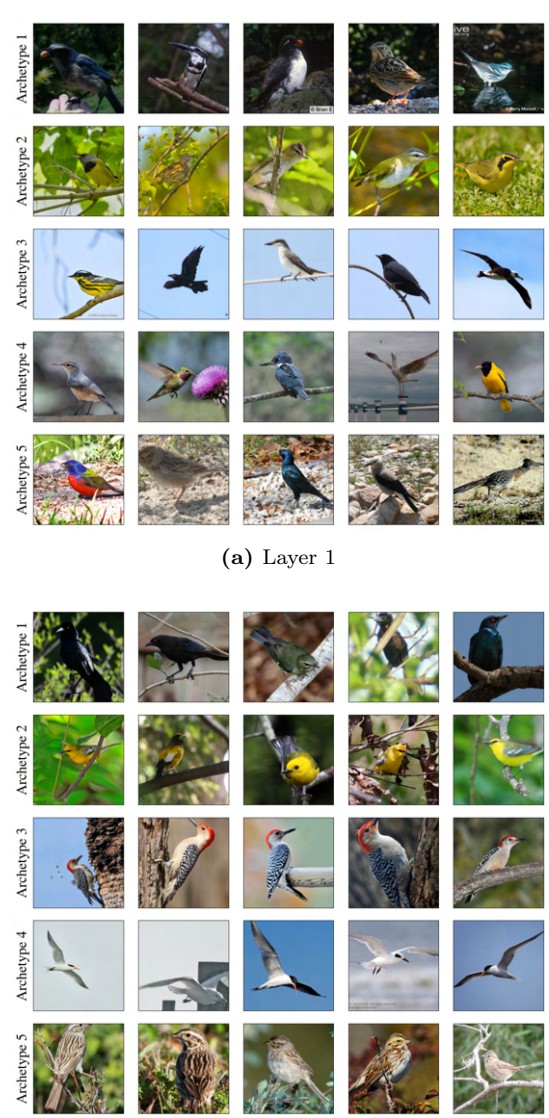

**(a)** Layer 1

**(b)** Layer 12

**Figure 3.** The five images closest to each archetype for the first and last layer for ViT trained on CUB.

archetype, vary with the different datasets, but the features defining each archetype and the qualitative interpretation of the archetypes remain constant with varying in-distribution datasets. In Figure 2(b), the consistency of archetypes for CUB for

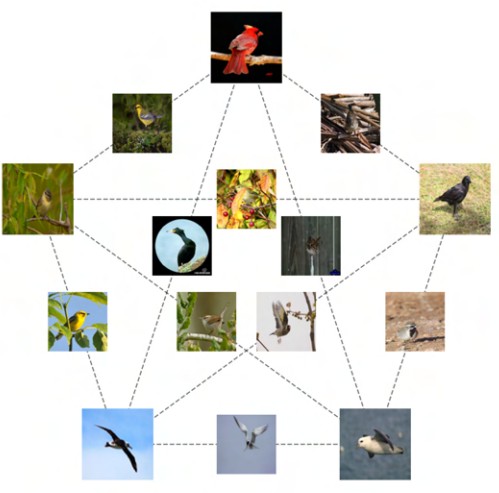

**Figure 4.** A visualization of the archetypal simplex of layer one with 5 archetypes for the ViT model. At the vertices the most archetypical images are shown for the representation and on each edge the images closest to the exact halfway point between the two archetypes are shown.

all three models is shown, the results for organCM-NIST can be seen in Figure D.1. We evaluate consistency both across repeated runs on the full dataset (baseline) and across different dataset splits. The results show near-perfect consistency across runs, indicating high stability of the AA solution. Importantly, consistency across dataset splits also remains high, exceeding 0.9 for most layers, highlighting that the archetypes are dataset independent and capture fundamental, model-driven structures in the latent space. These findings support the interpretation of archetypes as robust, data-independent descriptors of how models organize information in terms of extreme latent representations.

#### 4.2.1 Accuracy on the Archetypal Simplex

To evaluate how well the archetypal projection captures the predictive capacity of the latent space and to determine when class separation emerges in the network, we apply a KNN classifier to both the latent representations and their corresponding archetypal projections. As shown in Figure 2(c), classification accuracy increases progressively across layers, consistent with the expectation that deeper layers encode more task-relevant features. Interestingly, for ViT the accuracy of the KNN classifier is already reaching maximum accuracy after layer five, while DINOv2 and ResNet50 reach maximum accuracy only in the last layer. This can potentially be explained by the training procedure of these models. While ViT is pretrained in a supervised fashion,

which leads to more class-separated representations, DINOv2 is trained with a much larger data corpus in an unsupervised way, leading to more general representations. This could explain why DINOv2 only separated distinct classes in late layers due to the finetuning.

The classification accuracy on the archetypal simplex depends on the number of archetypes used to approximate the latent space. For OrganCMNIST, we employ 20 archetypes (Figure 2(c)), while for CUB we use 100 archetypes (Figure D.3). Across both datasets, the resulting accuracy is comparable to or in some cases higher than that obtained in the original latent space, demonstrating that substantial compression can be achieved with minimal loss of discriminative information. Notably, in the case of CUB, neither KNN applied to the archetypal simplex nor to the raw latent representations provides sufficient resolution to reliably classify bird species.

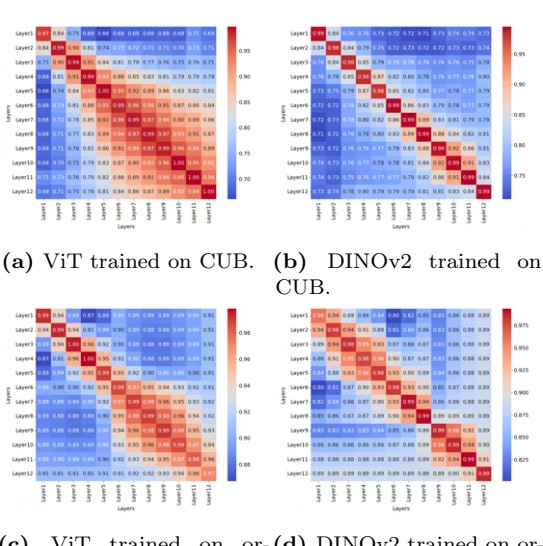

**(a)** ViT trained on CUB. **(b)** DINOv2 trained on CUB.

**(c)** ViT trained on organCMNIST. **(d)** DINOv2 trained on organCMNIST.

**Figure 5.** Layerwise comparison of archetypes for ViT and DINOv2 trained on CUB and organCMNIST reveals high similarities for blocks of layers in ViT and lower simiarity for DINOv2.

### 4.3 Layer-wise Comparison of Archetypes

We also investigate the similarity of archetypes across layers of the two transformer models to better understand how the latent space evolves and to identify fundamental differences in how each model processes the same data and task. We find a high degree of similarity in layers of the ViT (Figure 5(a) and 5(c)), which can be interpreted as high redundancy of information across layers, even across 3-4 layers, a phenomenon previously observed in transformer models [34]. The representations in the DINOv2 model change more rapidly from layer to layer

(Figure 5(b) and 5(d)), which could potentially be explained by the unsupervised pretraining with a much larger data set and, therefore, more diverse representations. Generally, the similarity across layers is higher for the simpler task (organCMNIST) than for the more complex task of CUB. The high similarity in later layers in ViT also matches with the high KNN accuracy already from layer five (Figure 2(c)), the model seems to capture class differences quite early and therefore doesn't need to build more distinct representations in later layers.

It is important to note that this comparison is based on five archetypes per layer. Although the optimal number of archetypes remains relatively stable across layers, this fixed number may slightly overestimate the true similarity between layers. Nonetheless, the overall trend remains consistent.

## 4.4   Explanation of Test Images

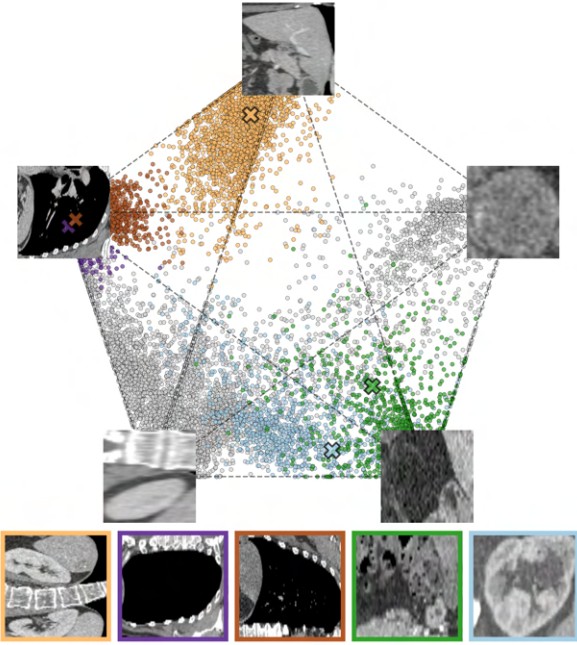

**Figure 6.** A simplex projection with five archetypes with their most archetypical images for the twelfth layer of the DINOv2 model for the OrganCMNIST data with unseen projected test images (bottom five images). The colored points on the simplex correspond to the class of the test images, marked by **X** on the simplex, the colors correspond to the frames of the test images. The grey points are projected test samples not belonging to one of the five highlighted classes.

Finally, we examine how new data points can be interpreted throughout the model using archetypal representations. In Figure 6, we project five test images onto the archetypal simplex, illustrating how each image can be expressed as a convex combination of five archetypes. This projection not only provides an interpretable decomposition of each test image

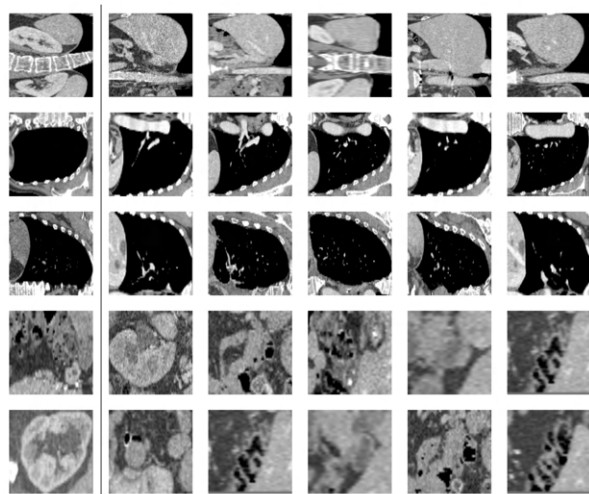

**Figure 7.** Five projected test images (left) and their five closest neighbors on the archetypal simplex (right)

but also reveals how well the archetypal simplex clusters different classes, as indicated by the class-colored points.

This analysis can be performed on all layers of the model, enabling a layer-wise interpretation of how individual data points are processed. As such, the full trajectory of a test image through the network can be described in terms of a small set of interpretable archetypes, offering a transparent view into the model's internal decision-making process.

In addition to projection-based explanations, we also explore example-based interpretations by identifying the closest training samples on the simplex for each test image. Examples of this approach are shown in Figure 7. This method provides insight into how new data points are positioned relative to the training distribution in the latent space and may help uncover potential biases or shortcut learning behaviors in the model.

## 5   Conclusion

To the best of our knowledge, we are the first to apply AA in a deep learning explainability context. We find that all layers can be represented with surprisingly high compression using the archetypal simplex with both high variance explained and NMI even for a low-dimensional simplex and high accuracy when using a KNN to classify on the resulting simplex. We show that the AA model, already recognized for its interpretability and intuitive structure, provides a powerful framework for analyzing the latent representations of neural networks. It reveals known issues such as over-parameterization, as demonstrated by the strong compression achievable and the KNN performance on the simplex, and it also uncovers non-trivial insights such as what the model prioritizes at each layer. Our framework therefore enables

both a global understanding of network behavior and local explanations of unseen test images at any stage of the model.

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

# A    Models and Data Details

The code for running the analysis and creating the figures can be found at github.com/Wedenborg/
Explaining-Latent-Representations-of-Neural-Networks-with-Archetypal-Analysis

We run all experiments on two different transformer-based vision models, namely ViT [26], DINOv2
[27] and on a ResNet50 [28]. All models were pretrained on variants of ImageNet [35] and retrieved from
hugginface.co.

We fine-tuned the models to perform classification of two different datasets, Caltech-UCSD Birds-200-
2011 Dataset [29] (CUB) and a dataset of the MedMNIST collection [30], the organcMNIST dataset
[31]. The trained models will be available on huggingface.co upon publication. All models were trained
with a batch-size of 32, a scheduler and early stopping for a maximum of 10000 steps, the learning rate
was optimized using a validation set. The exact learning rates and final test accuracies can be seen in
Table A.1.

**Table A.1.** Learning rate (lr) and test accuracy (acc) in % for all models trained on CUB and organcMNIST.

|  | CUB | | organcMNIST | |
| --- | --- | --- | --- | --- |
| model | lr | acc | lr | acc |
| ViT | $1e^{-4}$ | 0.84 | $1e^{-4}$ | 0.89 |
| DINOv2 | $1e^{-5}$ | 0.87 | $1e^{-5}$ | 0.90 |
| ResNet50 | $1e^{-4}$ | 0.76 | $1e^{-4}$ | 0.92 |

# B    Archetypal Analysis Results

In Table B.1 and Table B.2 we show the selection of the optimal number of archetypes for each layer of
each model and the NMI and VE for that number of archetypes for organCMNIST and CUB, respectively.

# C    Additional figures - NMI/VE and archetype examples

In Figure C.1 - C.4 the NMI and VE for all models, as well as 5 archetypes for all layers for the
organCMNIST are shown. For the CUB dataset, the same plots are shown in Figure C.5 - C.8.

# D    Additional figures - Miscellaneous

In Figure D.1, we report the archetype similarity across different network layers. To assess robustness, the
data were further partitioned into five independent subsets, and similarity was computed separately for
each split.
Figure D.2 presents the NMI between the simplex representation and the ground-truth labels. Finally,
Figure D.3 shows the KNN classification accuracy for the CUB dataset, comparing performance on the
simplex projection with that obtained from the raw latent representation.

**Table B.1.** Archetype evaluation for the OrganCMNIST dataset grouped by layer and model.

| Layer | Model | Best archetype count | NMI (mean ± std) | VE (mean ± std) |
|---|---|---|---|---|
| 1 | ResNet50 | 8 | $0.936 \pm 0.0133$ | $0.998 \pm 0.0001$ |
| | ViT | 5 | $0.920 \pm 0.0245$ | $0.979 \pm 0.0022$ |
| | DINOv2 | 4 | $0.881 \pm 0.0555$ | $0.978 \pm 0.0016$ |
| 2 | ResNet50 | 8 | $0.952 \pm 0.0129$ | $0.985 \pm 0.0004$ |
| | ViT | 5 | $0.954 \pm 0.0190$ | $0.939 \pm 0.0039$ |
| | DINOv2 | 7 | $0.930 \pm 0.0159$ | $0.960 \pm 0.0022$ |
| 3 | ResNet50 | 4 | $0.974 \pm 0.0321$ | $0.921 \pm 0.0006$ |
| | ViT | 5 | $0.963 \pm 0.0155$ | $0.875 \pm 0.0048$ |
| | DINOv2 | 10 | $0.939 \pm 0.0112$ | $0.968 \pm 0.0023$ |
| 4 | ResNet50 | 11 | $0.971 \pm 0.0098$ | $0.754 \pm 0.0053$ |
| | ViT | 5 | $0.952 \pm 0.0177$ | $0.825 \pm 0.0036$ |
| | DINOv2 | 8 | $0.933 \pm 0.0176$ | $0.945 \pm 0.0035$ |
| 5 | ViT | 10 | $0.964 \pm 0.0134$ | $0.826 \pm 0.0081$ |
| | DINOv2 | 8 | $0.944 \pm 0.0145$ | $0.922 \pm 0.0035$ |
| 6 | ViT | 11 | $0.981 \pm 0.0103$ | $0.847 \pm 0.0071$ |
| | DINOv2 | 4 | $0.971 \pm 0.0252$ | $0.862 \pm 0.0038$ |
| 7 | ViT | 11 | $0.978 \pm 0.0130$ | $0.865 \pm 0.0047$ |
| | DINOv2 | 9 | $0.974 \pm 0.0084$ | $0.899 \pm 0.0031$ |
| 8 | ViT | 11 | $0.979 \pm 0.0091$ | $0.882 \pm 0.0054$ |
| | DINOv2 | 8 | $0.953 \pm 0.0122$ | $0.865 \pm 0.0060$ |
| 9 | ViT | 12 | $0.975 \pm 0.0109$ | $0.902 \pm 0.0057$ |
| | DINOv2 | 9 | $0.959 \pm 0.0117$ | $0.850 \pm 0.0045$ |
| 10 | ViT | 11 | $0.969 \pm 0.0081$ | $0.898 \pm 0.0256$ |
| | DINOv2 | 9 | $0.961 \pm 0.0107$ | $0.823 \pm 0.0065$ |
| 11 | ViT | 13 | $0.981 \pm 0.0066$ | $0.939 \pm 0.0044$ |
| | DINOv2 | 8 | $0.963 \pm 0.0179$ | $0.804 \pm 0.0098$ |
| 12 | ViT | 11 | $0.978 \pm 0.0096$ | $0.923 \pm 0.0098$ |
| | DINOv2 | 8 | $0.953 \pm 0.0171$ | $0.768 \pm 0.0131$ |

**Table B.2.** Archetype evaluation for the CUB dataset grouped by layer and model.

| Layer | Model | Best archetype count | NMI (mean ± std) | VE (mean ± std) |
|---|---|---|---|---|
| 1 | ResNet50 | 3 | $0.891 \pm 0.0399$ | $0.925 \pm 0.0013$ |
|   | ViT | 4 | $0.970 \pm 0.0365$ | $0.789 \pm 0.0075$ |
|   | DINOv2 | 3 | $0.951 \pm 0.0330$ | $0.913 \pm 0.0021$ |
| 2 | ResNet50 | 3 | $0.846 \pm 0.0720$ | $0.909 \pm 0.0012$ |
|   | ViT | 5 | $0.947 \pm 0.0251$ | $0.785 \pm 0.0015$ |
|   | DINOv2 | 3 | $0.938 \pm 0.0388$ | $0.825 \pm 0.0028$ |
| 3 | ResNet50 | 3 | $0.863 \pm 0.0650$ | $0.864 \pm 0.0005$ |
|   | ViT | 5 | $0.940 \pm 0.0351$ | $0.776 \pm 0.0018$ |
|   | DINOv2 | 3 | $0.939 \pm 0.0391$ | $0.830 \pm 0.0017$ |
| 4 | ResNet50 | 4 | $0.900 \pm 0.0547$ | $0.443 \pm 0.0006$ |
|   | ViT | 3 | $0.980 \pm 0.0144$ | $0.746 \pm 0.0022$ |
|   | DINOv2 | 3 | $0.932 \pm 0.0443$ | $0.760 \pm 0.0021$ |
| 5 | ViT | 3 | $0.976 \pm 0.0188$ | $0.750 \pm 0.0016$ |
|   | DINOv2 | 4 | $0.936 \pm 0.0285$ | $0.749 \pm 0.0019$ |
| 6 | ViT | 6 | $0.898 \pm 0.0593$ | $0.763 \pm 0.0009$ |
|   | DINOv2 | 4 | $0.927 \pm 0.0287$ | $0.757 \pm 0.0024$ |
| 7 | ViT | 6 | $0.935 \pm 0.0240$ | $0.752 \pm 0.0011$ |
|   | DINOv2 | 5 | $0.948 \pm 0.0171$ | $0.820 \pm 0.0019$ |
| 8 | ViT | 6 | $0.921 \pm 0.0231$ | $0.706 \pm 0.0006$ |
|   | DINOv2 | 5 | $0.965 \pm 0.0149$ | $0.798 \pm 0.0013$ |
| 9 | ViT | 6 | $0.953 \pm 0.0302$ | $0.665 \pm 0.0009$ |
|   | DINOv2 | 7 | $0.959 \pm 0.0089$ | $0.844 \pm 0.0017$ |
| 10 | ViT | 6 | $0.932 \pm 0.0274$ | $0.621 \pm 0.0007$ |
|   | DINOv2 | 7 | $0.951 \pm 0.0207$ | $0.827 \pm 0.0018$ |
| 11 | ViT | 5 | $0.928 \pm 0.0254$ | $0.563 \pm 0.0009$ |
|   | DINOv2 | 6 | $0.935 \pm 0.0223$ | $0.814 \pm 0.0013$ |
| 12 | ViT | 5 | $0.956 \pm 0.0433$ | $0.332 \pm 0.0010$ |
|   | DINOv2 | 7 | $0.961 \pm 0.0301$ | $0.332 \pm 0.0002$ |

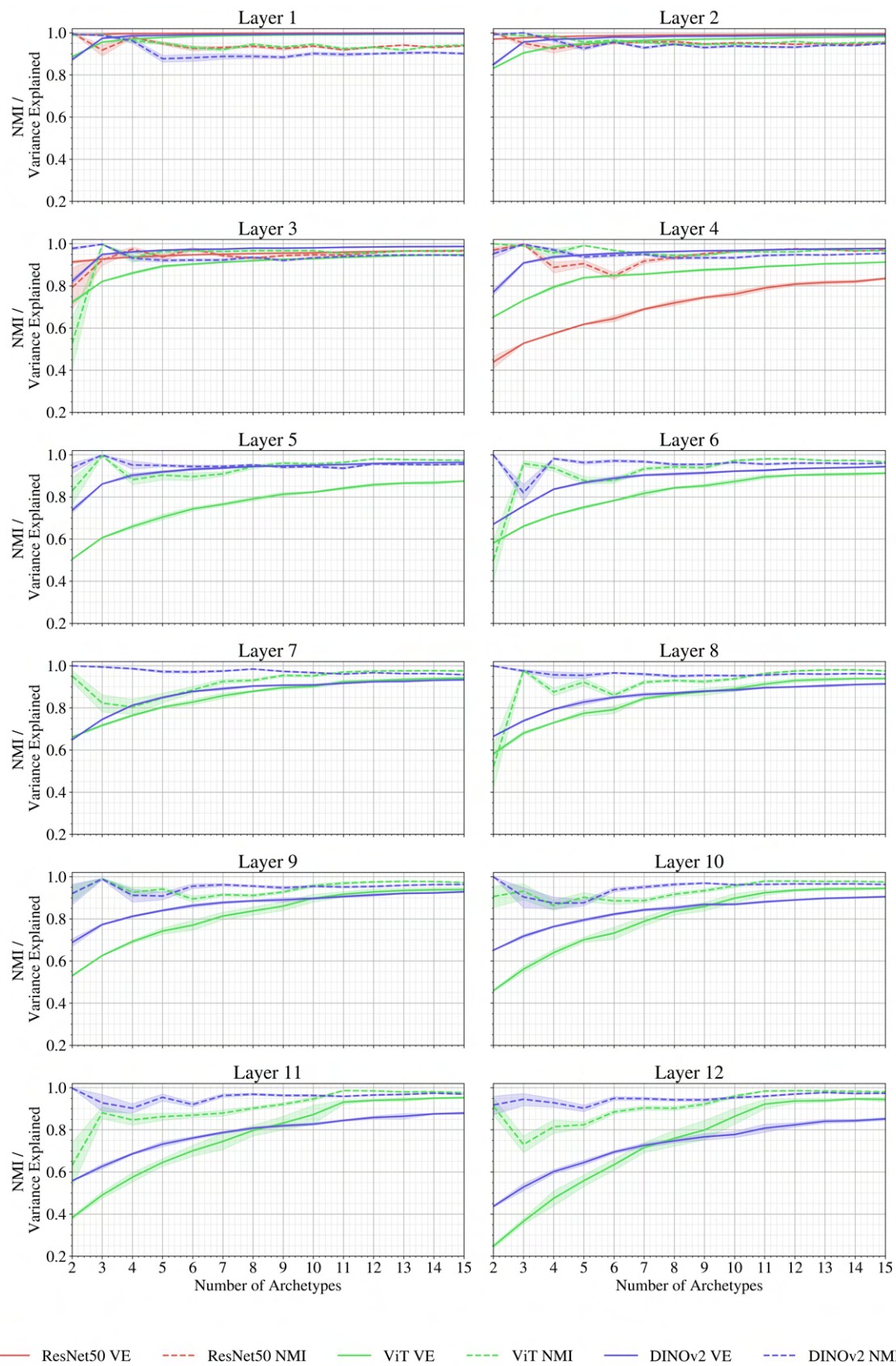

**Figure C.1.** Normalised Mutual Information and Variance explained across a different number of archetypes for the OrgancMNIST dataset

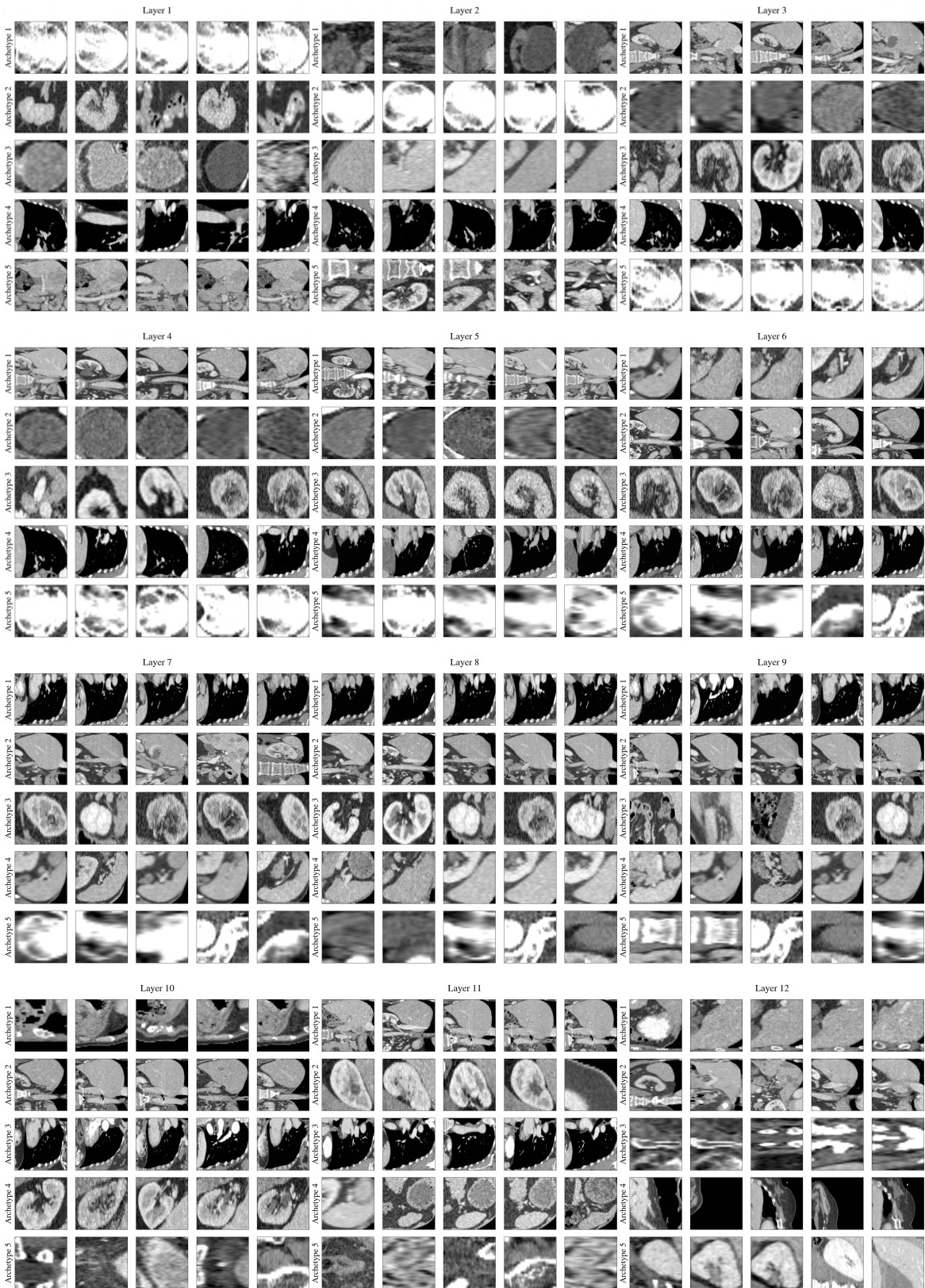

**Figure C.2.** The five images closest to the archetypes for all twelve transformer layers of the ViT model on organcmnist, for visualization purposes five archetypes has been chosen for all layers.

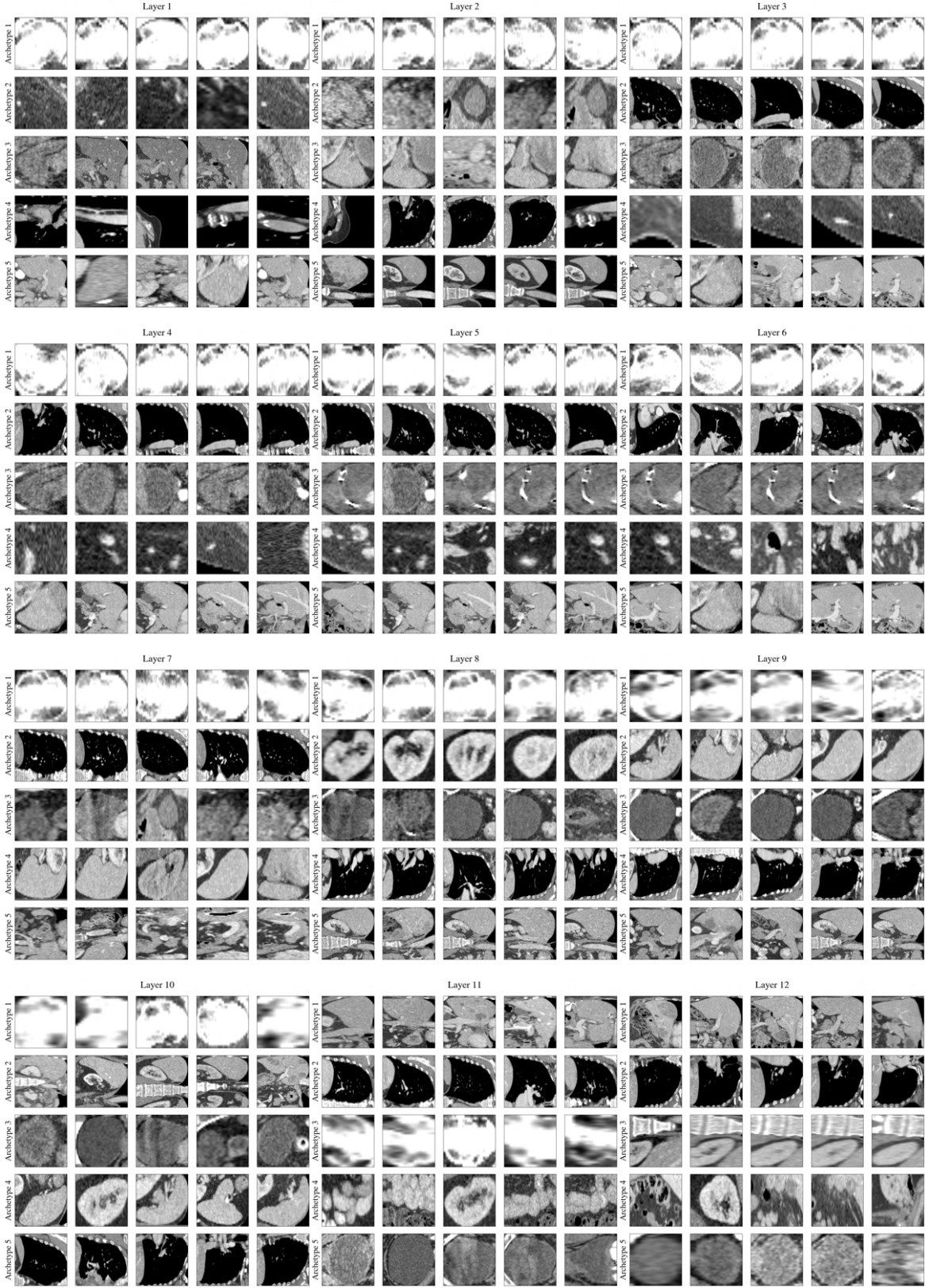

**Figure C.3.** The five images closest to the archetypes for all twelve transformer layers of the DINOv2 model on organcmnist, for visualization purposes five archetypes has been chosen for all layers.

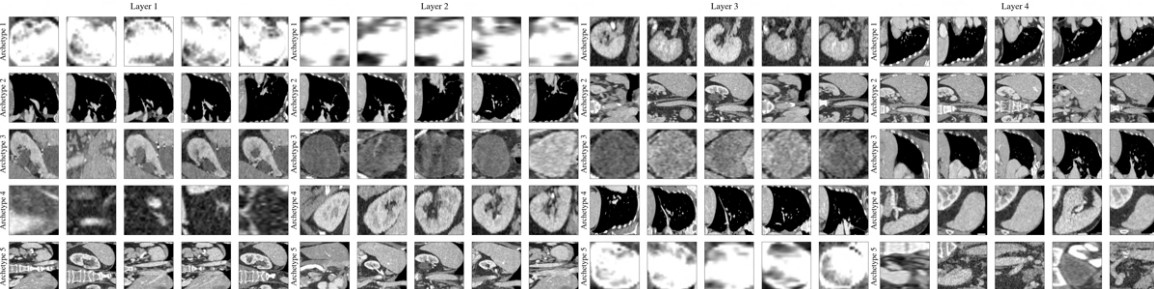

**Figure C.4.** The five images closest to the archetypes for the transformer layers of the ResNet50 model on *organcmnist*. For visualization, five archetypes are shown per layer.

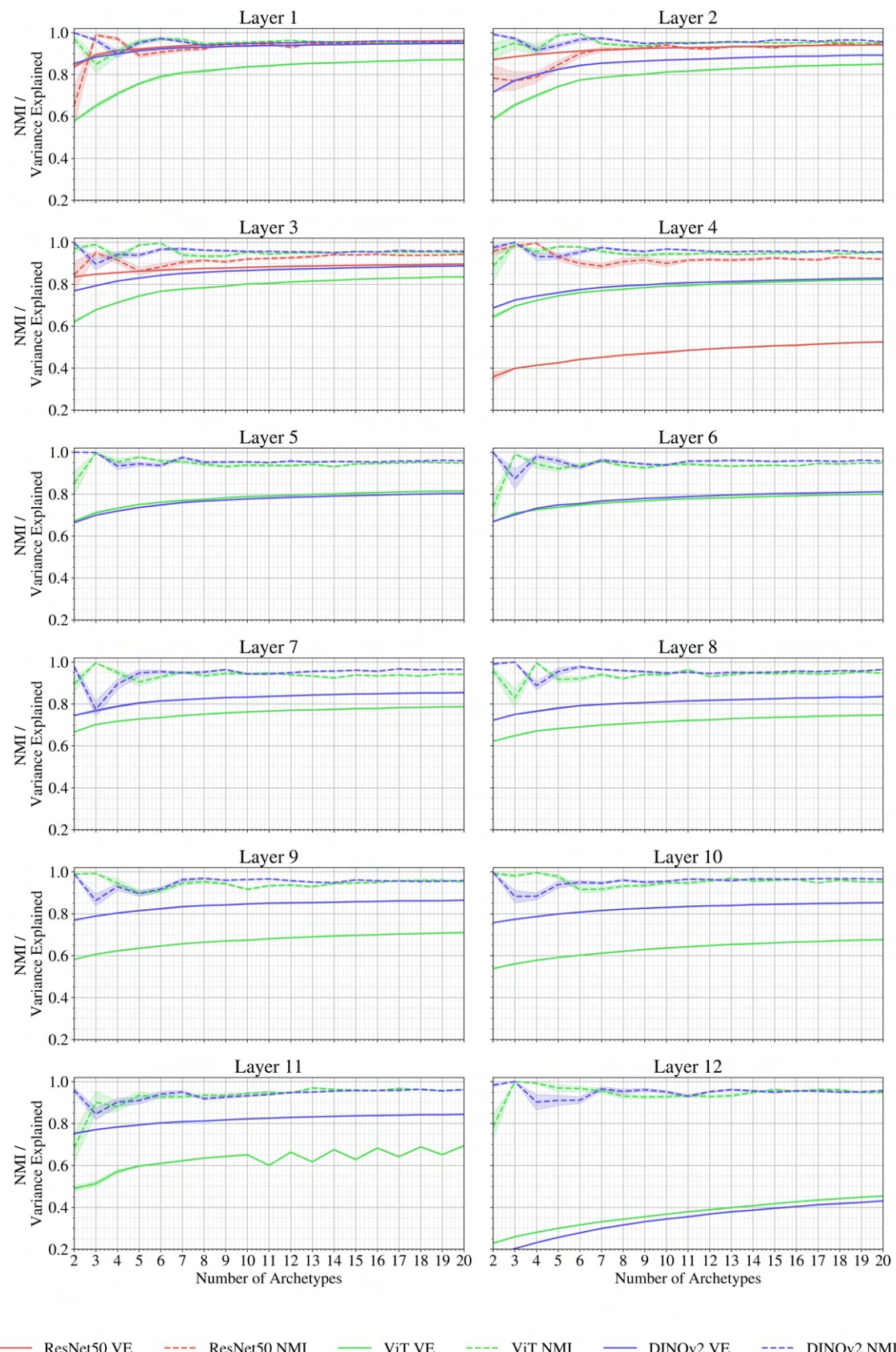

**Figure C.5.** Normalized Mutual Information and Variance explained across a different number of archetypes for the CUB dataset

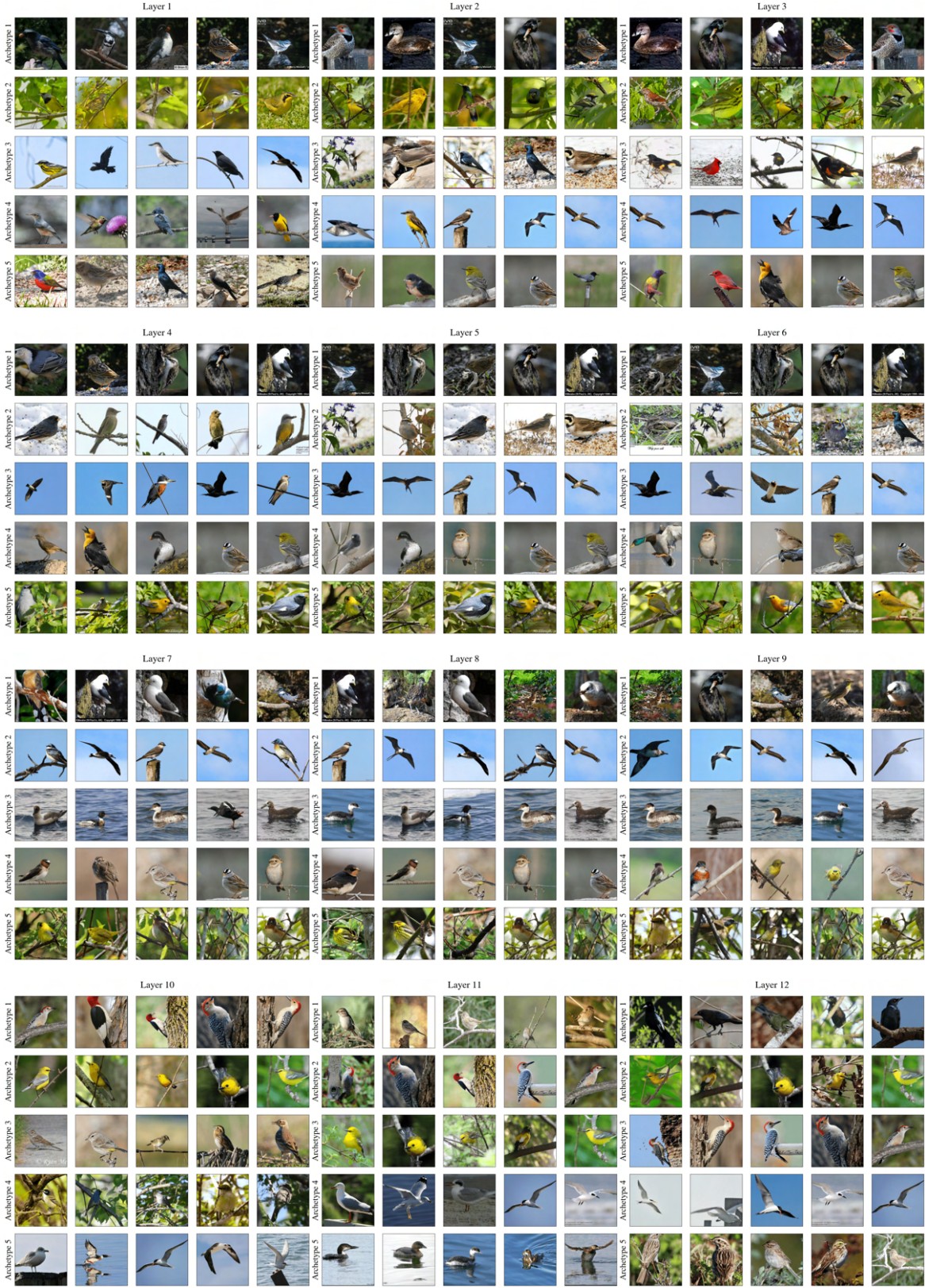

**Figure C.6.** The five images closest to the archetypes for all twelve transformer layers of the ViT model, for visualization purposes five archetypes has been chosen for all layers.

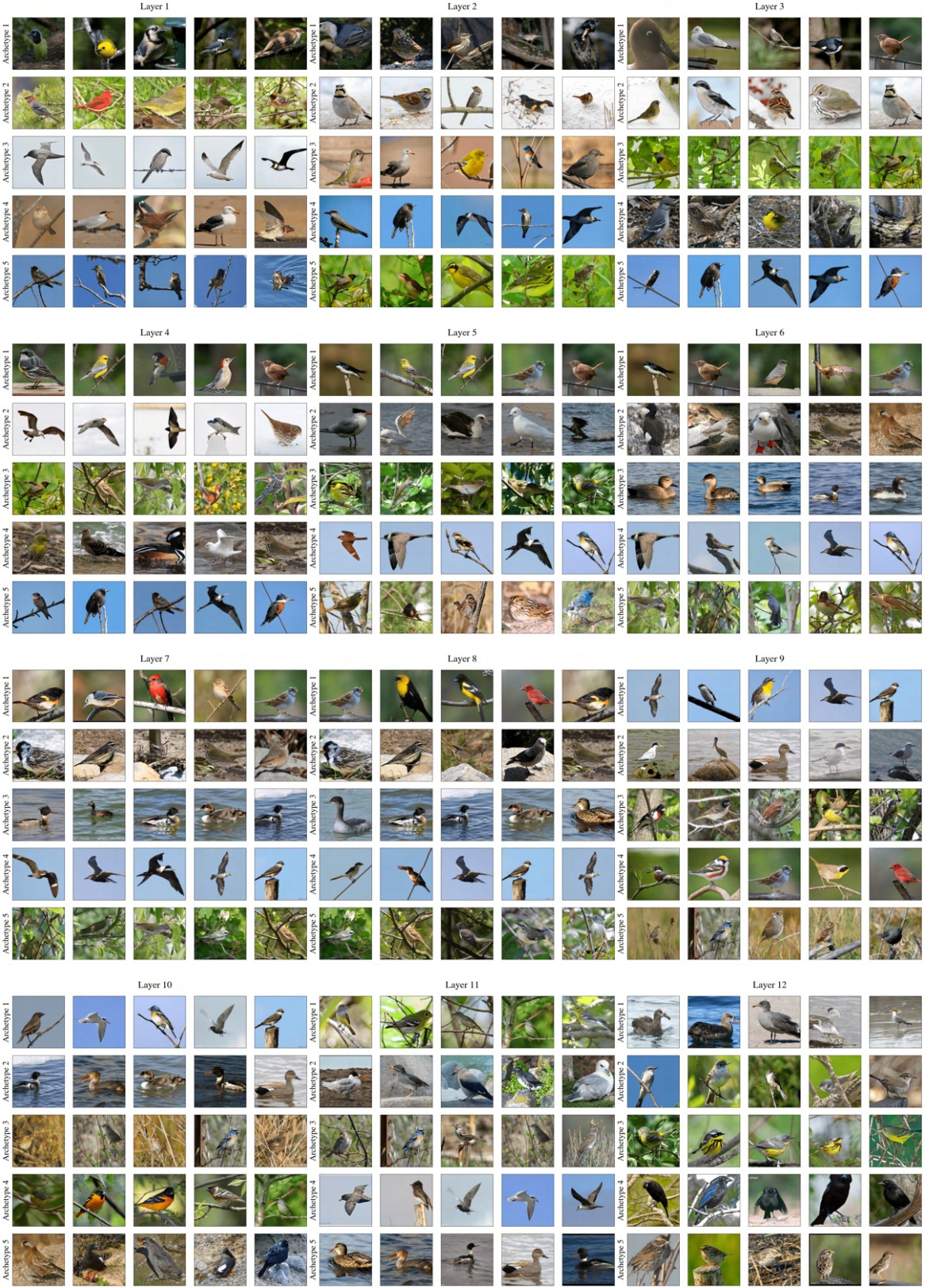

**Figure C.7.** The five images closest to the archetypes for all twelve transformer layers of the DINOv2 model, for visualization purposes five archetypes has been chosen for all layers.

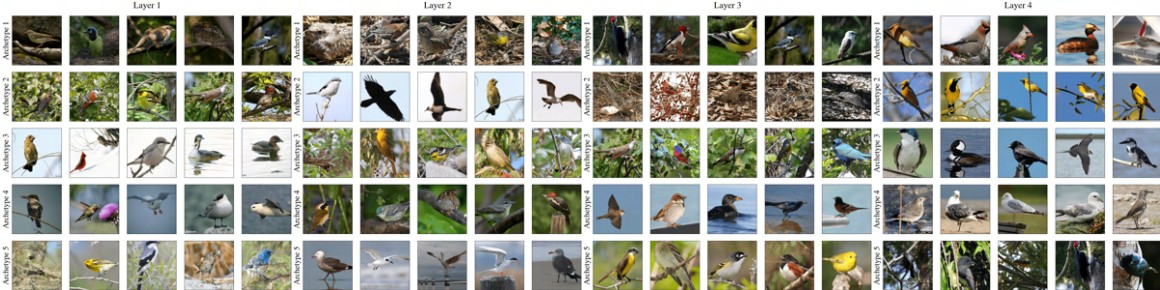

**Figure C.8.** The five images closest to the archetypes for the transformer layers of the ResNet50 model on *CUB*. For visualization, five archetypes are shown per layer.

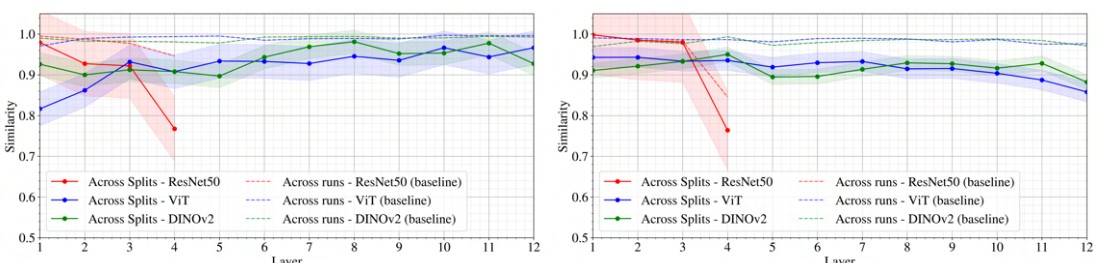

**(a)** Archetype consistency across splits and runs (baseline) for CUB.

**(b)** Archetype consistency across splits and runs (baseline) for organCMNIST.

**Figure D.1.** Archetype consistency.

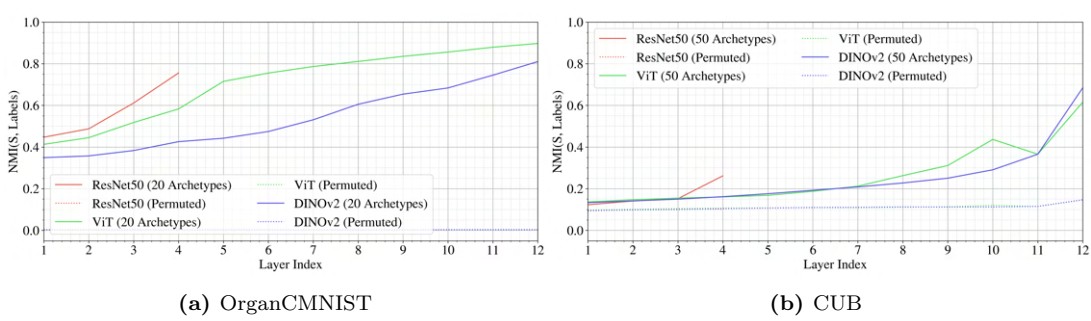

**(a)** OrganCMNIST

**(b)** CUB

**Figure D.2.** Normalized mutual information between the true labels and the archetypal simplex.

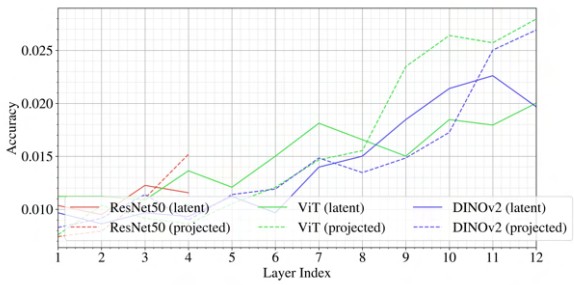

**Figure D.3.** K-nearest-neighbor accuracy for CUB

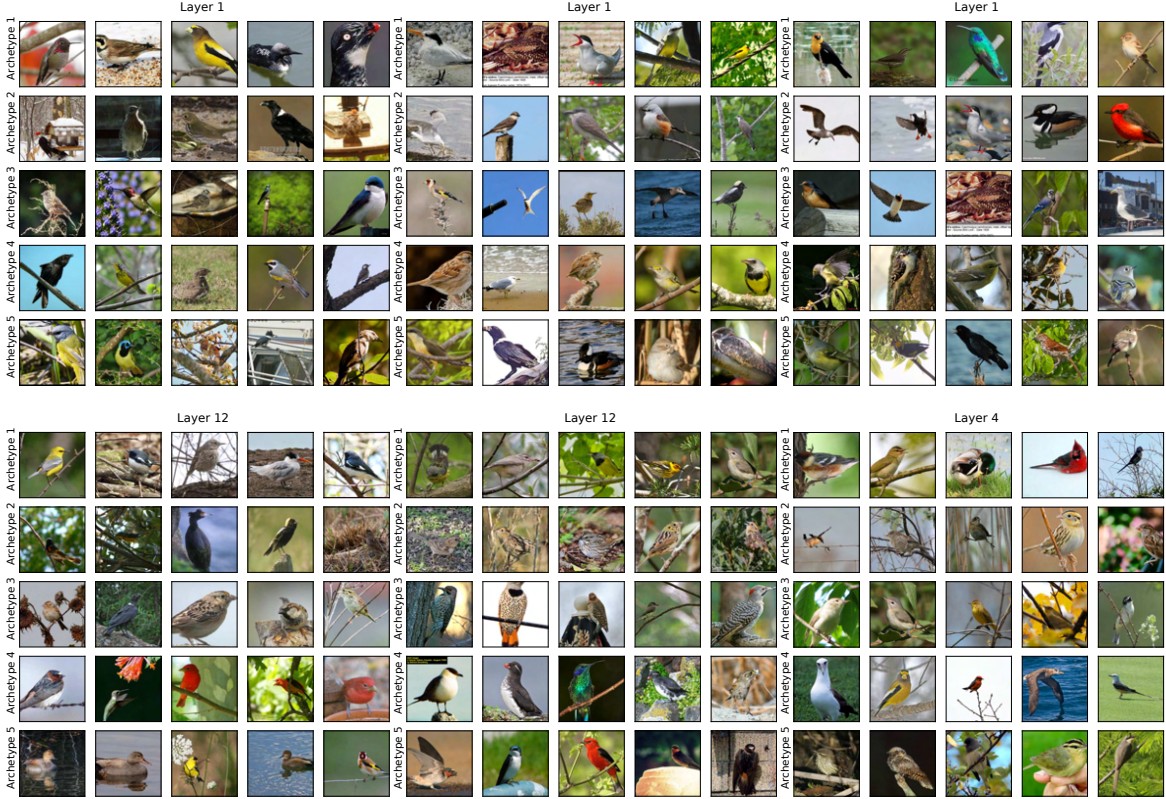

**Figure D.4.** Not-finetuned - non-normalized: The five images closest to the first and last archetype for the three models DinoV2 (left), ViT (middle) and ResNet50 (right), for visualization purposes five archetypes has been chosen for all layers.

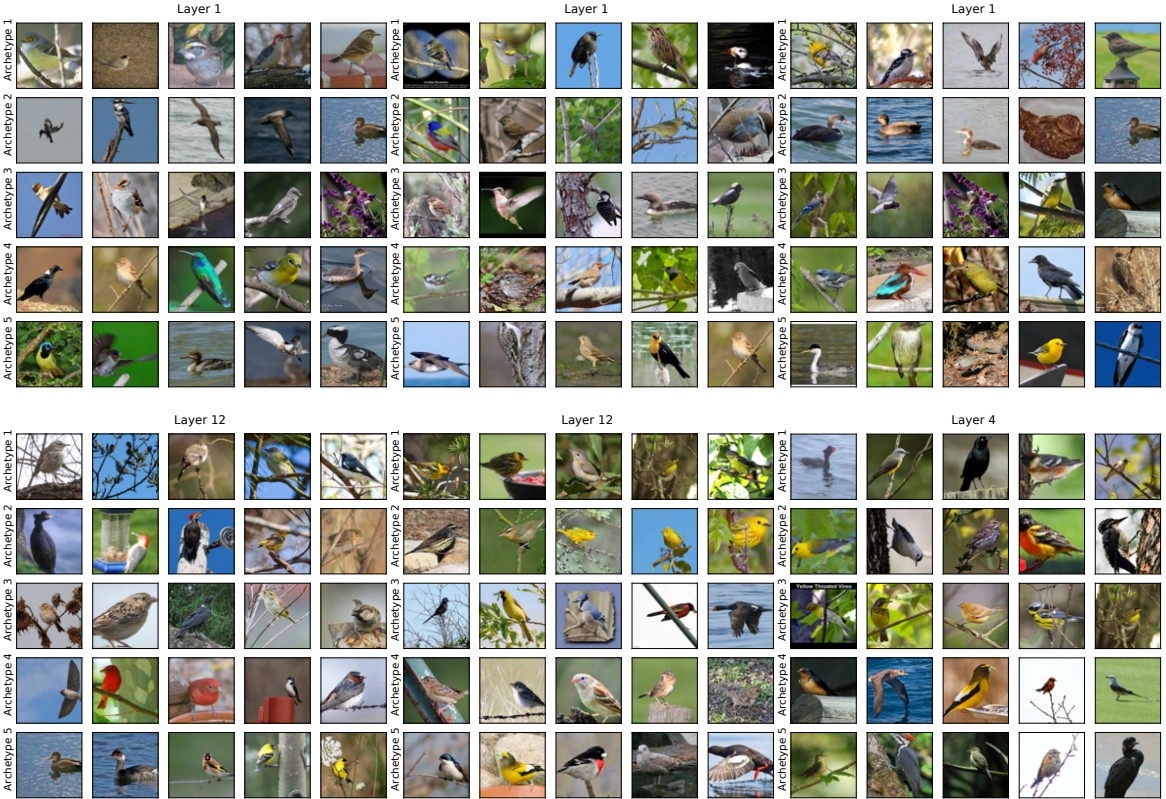

**Figure D.5.** Not-finetuned - normalized: The five images closest to the first and last archetype for the three models DinoV2 (left), ViT (middle) and ResNet50 (right), for visualization purposes five archetypes has been chosen for all layers.

