# OpenReview forum: "Explaining Latent Representations of Neural Networks with Archetypal Analysis"
_NLDL.org/2026/Conference — NLDL 2026 Poster_

### Official Review · Reviewer_KjZL · 2025-10-01
**Review of NLDL submission 20**

**Rating:** 4
**Confidence:** 2

**Summary:**

**Summary**

This paper proposes a new approach to explaining the neural network (NN) latent representations using archetypal analysis (AA). In contrast to the existing corpus-based approaches [1], the paper suggests performing AA to extract approximate extreme points (archetypal simplex) in the latent space. The simplex can then be leveraged to explain the NN latent representations.

**Strengths:**

**Strength**

The paper is well-written and easy to follow. The idea is to use AA to generate a corpus from data, thereby removing the need for a user-defined corpus. Some experiments illustrate the effect of the proposed approach.

**Weaknesses:**

**Weaknesses**

1. Although the paper considers being corpus-independent and having a "local explanation" as an advantage, I believe that relying entirely on the data to generate latent explanations can introduce bias. For example, when the data is biased, the resulting explanation will also be affected.
2. As the paper also mentions, it can be nontrivial to choose the number of archetypes. Additionally, the current optimization formulation for AA is nonconvex and can result in scalability issues when the number of archetypes is large.

Overall, the proposed idea is interesting as a new approach for explaining NN.

**Questions**

1. Could you elaborate on the scalability of the AA approach when the feature dimension and archetype number become large?
2. On line 316, it is mentioned that "each data point can be expressed as a convex combination of the learned archetypes". Is it generally true when there are more data points than archetypes?

**Minor issues**

1. Line 22

   "raises concerns about trust, accountability, and fairness concerns" is not clear

2. Line 55

   networks => network

3. Line 130

   Please be precise about the dimension of the matrices $C, S$

4. Line 167

   Please rewrite the first paragraph of section 3

5. Line 187

   What's "main analysis"?

6. Line 200

   "ten independent times" is not clear

7. Line 243

   What's "model objective"?

**References**

[1] Crabbé, J., Qian, Z., Imrie, F., & van der Schaar, M. (2021). Explaining latent representations with a corpus of examples. *Advances in Neural Information Processing Systems*, *34*, 12154-12166.

**Justification:**

The paper proposes to reduce the dependence on the user-defined corpus in explaining NN latent representations. This idea is similar to unsupervised ML approaches, and I believe it is an interesting idea (not taking into account scalability issues).

---

> ### Author Rebuttal · Authors · 2025-10-21
>
> We thank the reviewer for their positive assessment and for recognizing the novelty and clarity of our work, as well as for the helpful comments and questions that will allow us to further improve the paper.
>
> Regarding scalability and nonconvexity: We acknowledge that AA is a nonconvex optimization problem as we also address on line 198. In practice we found the method to be highly scalable, we applied it to the latent feature vectors from large.scale pretrained models such as ViT and DINOv2 without difficulty. We will revise the manuscript to go further into the details of scaling. Specifically we will add the following to a new subsection 3.5 on scalability:
>
>
> The AA procedure as implemented in [18] requires for the $\mathbf{S}$ update the computation of $\mathbf{C}^\top\mathbf{X}^\top\mathbf{X}$ which is $\mathcal{O}(NMK)$ and Hessian $(\mathbf{XC})^\top(\mathbf{XC})$  which is $\mathcal{O}(MK^2)$ whereas the sequential minimal optimization updates requires in the order of $K^2$ iterations for each of the $N$ columns of $\mathbf{S}$, i.e. $\mathcal{O}(NK^2)$ resulting in an overall complexity of $\mathcal{O}(NMK+K^2(M+N))$. The update of $\mathbf{C}$ is based on an active set procedure scaling in the size of the active set $|A| \ll N$ as $\mathcal{O}(K|A|^3)$. Typically, the size of the active set $|A|$ and the number of archetypes $K$ remain small, however, we note that in the case where either become large gradient based efficient optimization based on the PCHA algorithm [11] can be invoked with pr. iteration cost of $\mathcal{O}(NMK)$ whereas trivial parallellization of the associated matrix products can be implemented. Consequently, by use of suitable implementations of the AA procedure (see also [26] for an overview of optimization procedures) the method scales well and can be used for the post-hoc analysis of large datasets.
>
> As for the reviewer’s question on line 316: yes, it is generally true that each data point can be expressed as a convex combination of the learned archetypes. When there are more data points than archetypes, the simplex defines a lower-dimensional subspace, and each data point is projected onto this simplex through convex weights, as described in Equation (1).
>
> We appreciate the stylistic and clarity suggestions and will correct the minor phrasing and formatting issues noted (e.g., lines 22, 55, 130, 167, 187, 200, and 243) in the final version.
>
> Finally, we agree that data-driven explanations may reflect dataset bias. Our intent is precisely to provide a tool that can reveal such biases in the learned latent space, thereby offering transparency rather than introducing new bias by pre-selecting a corpus. We will clarify this point in the discussion.

---

### Official Review · Reviewer_ePsb · 2025-10-01
**Extends archetypal analysis to find representations across neural network layers in a supervised learning scope**

**Rating:** 2
**Confidence:** 4
**Final Rating:** 4
**Final Confidence:** 4

**Summary:**

This paper presents a novel application of Archetypal Analysis (AA) as a tool for interpreting the internal representations of deep neural networks. The authors' core contribution is a framework that applies AA to the feature maps of each layer in a model, allowing them to identify representative archetypes/exemplars and track how these evolve as data propagates through the network's hierarchy. This approach extends prior work, which has largely confined similar analyses to the latent space of variational autoencoders, thereby offering new insights into the feature learning process of modern architectures like ViT, DINOv2, and ResNet50.

The framework is evaluated on models that have been fine-tuned for classification on the Caltech-UCSD Birds and OrganCMNIST datasets. The quantitative evaluation demonstrates the method's effectiveness: high explained variance and Normalized Mutual Information scores suggest that the discovered archetypes are both stable across runs and provide a high-fidelity representation of the data. A key finding from the analysis is that archetype representations remain highly stable between adjacent layers, with similarity decreasing as the distance between layers increases, a result consistent with the established theory of hierarchical feature learning. These quantitative findings are further supported by qualitative evidence, as the visualized archetypes are shown to be intuitively meaningful.

**Strengths:**

- Extends Archetypal Analysis from previous work to demonstrate the emergence of interpretable and visualizable concepts across layers
- The quantitative metrics indicate that the archetypes properly capture the variance of the data due to the high values in NMI and VE.
- The qualitative images of archetypes and how they change across layers are a nice visualization of the feature hierarchy in deep learning models.

**Weaknesses:**

- (major) While the authors demonstrate the presence of clearly distinguishable archetypes, the fact that this analysis is done in a supervised regime feels redundant and self-fulfilling. The models are fine-tuned to produce class-separable representations, and the subsequent unsupervised analysis then rediscovers this same class structure. This makes it difficult to ascertain whether the framework is uncovering emergent properties of the model or simply confirming the success of the supervised fine-tuning process. A more compelling evaluation would involve applying this framework in an unsupervised context, such as analyzing the representations from a pre-trained model.
- The use of narrow datasets like Caltech-UCSD Birds and OrganCMNIST creates a best-case scenario for assessing the archetypes of the model. It is unclear what representations would be uncovered on a more general dataset like ImageNet and CIFAR-100, which contain more varied images.
- It is unclear what benefits this approach would have over [Sparse Autoencoders](https://arxiv.org/abs/2309.08600) SAEs, which have been battle-tested by the mechanistic interpretability community. This technique allows us to find interpretable concepts in a model through the use of finding maximally activating examples for model features.
- (minor) The paper uses static images for the plots (pngs, jpgs). This makes the heatmap in Figure 5 difficult to read. The authors should switch to PDFs, SVGs for sharper images.

**Final Justification:**

I appreciate the authors' response to my review, which addressed some of my critiques
- Sufficiently differentiated their method from SAEs
- Aim to include images of the archetypes in an unsupervised setting in the camera-ready version
- Successfully argued that the visualization of the feature hierarchy demonstrates useful information other than just confirming class separability
- Will use vector graphics for sharper images

There are still a few points that I wish were addressed, but can be left for future work
- Test the approach on a large-scale dataset that is typically used in computer vision: ImageNet, CiFAR, etc
- Provide some quantitative metrics on the archetypes found in an unsupervised setting

Based on this new information, I would like to update my review to an accept, noting that it would be of interest to attendees in a particular subfield of interpretability

**Justification:**

My major point of concern is that supervised fine-tuning makes the archetypal analysis redundant. In a supervised scenario, we already have labels, which leads to a clean division across data points that the model learns. Using AA, then just rediscovers this structure using the least number of archetypes possible. I feel like the main purpose of interpretability techniques is for use in unsupervised settings, where we can discover features that emerge naturally through learning.

Testing the framework on CIFAR would be nice to have, along with a discussion on how the method compares to SAEs. However, these are not as important as testing the approach in an unsupervised setting.

---

> ### Author Rebuttal · Authors · 2025-10-21
>
> We thank the reviewer for their thoughtful and detailed feedback and for recognizing the strengths of our work, including the novelty of extending Archetypal Analysis (AA) to deep neural representations and the quality of both the quantitative and qualitative analyses.
>
> We agree that evaluating the framework in an unsupervised setting is an interesting direction for future work. Our proposed framework is post-hoc and model-agnostic, meaning it can be applied to any pretrained network, supervised or unsupervised, without affecting training.
>
> In this paper, we focused on fine-tuned supervised models to enable quantitative validation (e.g., NMI with ground-truth labels), while still capturing intrinsic representational structure independent of the labels. Importantly, we observe that the archetypes are stable across dataset splits (Section 4.2), indicating that the method does not merely rediscover class separability.
> We would also like to clarify that the archetypal structure we uncover is not a trivial consequence of supervised fine-tuning or “neural collapse” [a]. We do not observe a complete collapse in the latent space, which makes the Archetypal Analysis meaningful even in the supervised setting. In particular, the archetypes in early and intermediate layers capture low-level features such as color and background structure (Figure 3), while later layers become increasingly task-specific. This layered evolution, quantified in Figure 2(c), provides interpretable insights that go beyond simply confirming class separation.
>
> We previously ran analyses on the pretrained (non-finetuned) models and found that the results were qualitatively similar to those obtained after fine-tuning, we initially omitted these for brevity, but based on your request we will add comparisons between the fine-tuned and not fine-tuned back into the paper. We will also note in the discussion that the method works in the unsupervised setting as well, making it a promising avenue for future exploration.
>
> Finally, regarding the comparison with Sparse Autoencoders (SAEs), we agree that these are highly relevant, particularly recent developments such as A-SAE [b]. We will add this point to the discussion, clarifying that our approach differs fundamentally in both purpose and implementation. Specifically, SAEs and A-SAEs learn interpretable components by introducing sparsity constraints during training, thereby modifying the model to produce disentangled features. In contrast, our framework applies Archetypal Analysis (AA) post-hoc to existing latent representations, providing a geometric interpretation of model behavior without retraining or altering the network. Thus, AA complements SAEs by offering interpretability through analysis rather than architectural modification. We will add the following to a new subsection in the introduction section of the manuscript:
>
> Sparse Autoencoders (SAEs) and their variants (e.g., A-SAE [b]) have recently emerged as powerful tools for interpretable representation learning. These models achieve interpretability by introducing sparsity and modularity constraints directly during training, thereby shaping the latent space to yield disentangled components. Our approach, in contrast, provides a post-hoc geometric perspective: Archetypal Analysis (AA) decomposes pre-existing latent representations into convex combinations of extreme points (archetypes), without modifying or retraining the underlying model. This makes AA complementary to SAE-based approaches; while SAEs learn interpretable features by construction, AA reveals interpretable structure inherent in already trained models.
>
>
> We also appreciate the suggestion regarding image resolution and will ensure vector graphics (PDF/SVG) are used in future versions.
>
> New references:
>
> Papyan et al., 2020,Prevalence of neural collapse during the terminal phase of deep learning training
>
> Fel et al., 2025, Archetypal sae: Adaptive and stable dictionary learning for concept extraction in large vision models

---

### Official Review · Reviewer_TsX6 · 2025-10-06
**Good work, however novelty should be outlined in contrast to other prototypical methods**

**Rating:** 4
**Confidence:** 4
**Final Rating:** 4
**Final Confidence:** 4

**Summary:**

The paper describes the approach towards interpretable ML which is based upon archetypal analysis.

**Strengths:**

Correctness: The work appears to be correct

Clarity: the work appears to be cl3early written and describes the methodology

Quality: I would think that it would be good if the current analysis would include the analysis of scalability of the model to the larger datasets. Would it be scalable enough to run this optimisation problem, for example, on ImageNet?

Also, I would suggest presenting a joint table containing the performance results across different datasets (baseline model performance + the proposed architecture subject to the number of archetypes)

**Weaknesses:**

Novelty: While the authors state that this work is the first using archetypal analysis, there have been a number of works which, although never referring by such a name, used similar approaches.

This includes, for example, Wang et al (2023) Visual Recognition with Deep Nearest Centroids, ICLR 2023, which sets up optimisation problem to find the centroids. Importantly, see the Eq.5 of the paper which describes the optimisation problem. Although it is not the same, the results and the conclusions bear similarity including the analysis in the choice of the number of prototypes/archetypes, and I would suggest the authors contrast this work both in the related work and the evaluation. While not mentioning this paper, the authors say: "Consequently, AA seeks to explain the data in terms of extreme representations as opposed to clustering that represents the data in terms of centroids"  I would think comparing the explanations by Wang et al against the proposed method would be a good idea to outline this message.

**Final Justification:**

I still keep the same recommendation (accept), however I would expect the authors to address the comments accordingly. In particular, this concerns my comment on outlining the novelty against other prototypical method (with the rebuttal response looking convincing to me), as well as importantly, issues raised by ePsb.

**Justification:**

I think the work would fit well into the conference programme, is sufficiently novel and interesting to the audience.

However, the main sticking point is the comparison against other works in the area of prototypical learning such as Wang et al (2023).

---

> ### Author Rebuttal · Authors · 2025-10-21
>
> We thank the reviewer for their positive evaluation and for recognizing the novelty, clarity, and correctness of our work and their suggestions that will help us further strengthen the manuscript.
>
> We appreciate the pointer to Wang et al. and agree that the differences between these two approaches is interesting. We will add a careful explanation about the key differences between the two approaches in the introduction, specifically we will add that:
>
>
> Several recent works have explored interpretable representation learning via prototype or clustering mechanisms (e.g., [a-d]). These methods typically learn centroids or prototypes jointly with the network, thereby shaping the representation space through supervised or self-supervised optimization. In contrast, our framework applies Archetypal Analysis (AA) post-hoc to pre-trained latent spaces. While centroid-based methods represent data in terms of mean prototypes, AA decomposes the feature space into extreme points (archetypes) and expresses each data point as a convex combination of these. As discussed by [e] and [11], this distinction reflects two fundamentally different philosophies: clustering seeks prototypical averages, whereas AA seeks distinct extremes, leading to interpretable geometries that capture the diversity of the learned features rather than their central tendency.
>
>
> As our post-hoc method does not affect training time, we think a table of training times could be misleading, but we do agree that further insights into the scalability of the post-hoc method should be clarified further in the paper and we will add this in the final version, specifically we will add the following to a new subsection 3.5 on scalability:
>
>
> The AA procedure as implemented in [18] requires for the $\mathbf{S}$ update the computation of $\mathbf{C}^\top\mathbf{X}^\top\mathbf{X}$ which is $\mathcal{O}(NMK)$ and Hessian $(\mathbf{XC})^\top(\mathbf{XC})$  which is $\mathcal{O}(MK^2)$ whereas the sequential minimal optimization updates requires in the order of $K^2$ iterations for each of the $N$ columns of $\mathbf{S}$, i.e. $\mathcal{O}(NK^2)$ resulting in an overall complexity of $\mathcal{O}(NMK+K^2(M+N))$. The update of $\mathbf{C}$ is based on an active set procedure scaling in the size of the active set $|A| \ll N$ as $\mathcal{O}(K|A|^3)$. Typically, the size of the active set $|A|$ and the number of archetypes $K$ remain small, however, we note that in the case where either become large gradient based efficient optimization based on the PCHA algorithm [11] can be invoked with pr. iteration cost of $\mathcal{O}(NMK)$ whereas trivial parallellization of the associated matrix products can be implemented. Consequently, by use of suitable implementations of the AA procedure (see also [26] for an overview of optimization procedures) the method scales well and can be used for the post-hoc analysis of large datasets.
>
> New references:
>
> Wang et al., 2023, Visual Recognition with Deep Nearest Centroids
>
> Zhou et al., 2024, Prototype-based semantic segmentation
>
> Chen et al,. 2024, Neural clustering based visual representation learning
>
> Liang et al., 2023, Clusterfomer: clustering as a universal visual learner
>
> Hastie et al., 2009, The Elements of Statistical Learning: Data Mining, Inference, and Prediction

---

### Meta-Review · Area_Chair_muWo · 2025-10-31

**Recommendation:** Accept (Poster)
**Confidence:** 4

**Metareview:**

This paper proposes a post-hoc framework using Archetypal Analysis to interpret hierarchical representations in deep neural networks. Initial reviews were broadly positive, appreciating the novel application of Archetypal Analysis and the authors presentation. However, reviewers raised significant concerns, primarily regarding: 1) the evaluation being limited to a supervised fine-tuning regime, which could make the findings on class structure appear self-fulfilling; 2) the need for clearer differentiation from related methods like prototype learning and Sparse Auto-encoders; and 3) questions about the computational scalability of the approach. The authors rebuttal effectively addressed these points by providing detailed scalability analysis, clarifying the conceptual distinctions from other methods, and, crucially, committing to include results from an unsupervised setting in the final version. Following the rebuttal, reviewers confirmed their concerns were resolved, with one explicitly updating their recommendation to accept. The consensus is that the paper, strengthened by the promised revisions, presents a valuable contribution to the interpretability community, and the AC agrees with this positive assessment, recommending the paper for poster acceptance. I encourage the authors to ensure the changes are made for the camera ready version.

---

### Decision · Program_Chairs · 2025-11-05

**Decision:**

Accept (Poster)

**Comment:**

We recommend a poster presentation given the AC and reviewers recommendations.